# Cardiac MRI: An Alternative Method to Determine the Left Ventricular Function

**DOI:** 10.3390/diagnostics13081437

**Published:** 2023-04-17

**Authors:** Kerstin Michler, Christopher Hessman, Marcus Prümmer, Stephan Achenbach, Michael Uder, Rolf Janka

**Affiliations:** 1Faculty of Medicine, Friedrich-Alexander-University Erlangen-Nürnberg (FAU), 91054 Erlangen, Germany; 2Institute of Radiology, University Hospital Erlangen, Friedrich-Alexander-Universität Erlangen-Nürnberg, 91054 Erlangen, Germany; 3Chimaera GmbH, 91054 Erlangen, Germany; 4Department of Medicine 2–Cardiology and Angiology, Friedrich-Alexander-University Erlangen-Nürnberg (FAU), Universitätsklinikum, 91054 Erlangen, Germany

**Keywords:** cardiac MRI, left ventricular function, ejection fraction, MRI, cine MRI

## Abstract

(1) Background: With the conventional contour surface method (KfM) for the evaluation of cardiac function parameters, the papillary muscle is considered to be part of the left ventricular volume. This systematic error can be avoided with a relatively easy-to-implement pixel-based evaluation method (PbM). The objective of this thesis is to compare the KfM and the PbM with regard to their difference due to papillary muscle volume exclusion. (2) Material and Methods: In the retrospective study, 191 cardiac-MR image data sets (126 male, 65 female; median age 51 years; age distribution 20–75 years) were analysed. The left ventricular function parameters: end-systolic volume (ESV), end-diastolic volume (EDV), ejection fraction (EF) and stroke volume (SV) were determined using classical KfW (syngo.via and cvi42 = gold standard) and PbM. Papillary muscle volume was calculated and segmented automatically via cvi42. The time required for evaluation with the PbM was collected. (3) Results: The size of EDV was 177 mL (69–444.5 mL) [average, [minimum–maximum]], ESV was 87 mL (20–361.4 mL), SV was 88 mL and EF was 50% (13–80%) in the pixel-based evaluation. The corresponding values with cvi42 were EDV 193 mL (89–476 mL), ESV 101 mL (34–411 mL), SV 90 mL and EF 45% (12–73%) and syngo.via: EDV 188 mL (74–447 mL), ESV 99 mL (29–358 mL), SV 89 mL (27–176 mL) and EF 47% (13–84%). The comparison between the PbM and KfM showed a negative difference for end-diastolic volume, a negative difference for end-systolic volume and a positive difference for ejection fraction. No difference was seen in stroke volume. The mean papillary muscle volume was calculated to be 14.2 mL. The evaluation with PbM took an average of 2:02 min. (4) Conclusion: PbM is easy and fast to perform for the determination of left ventricular cardiac function. It provides comparable results to the established disc/contour area method in terms of stroke volume and measures “true” left ventricular cardiac function while omitting the papillary muscles. This results in an average 6% higher ejection fraction, which can have a significant influence on therapy decisions.

## 1. Introduction

The cardiac function parameters end-diastolic volume (EDV) and end-systolic volume (ESV) can be measured using MRI in addition to echocardiography [1,2,3,4], with MRI being superior to transthoracic echocardiography [5,6,7]. For this purpose, SSFP (steady state free precision) cine sequences are scanned as short axis stacks and evaluated with the contour surface method (KfM). In this method, the areas enclosed by endocardium on each acquired slice plane are determined during end-diastole and end-systole and multiplied by the slice thickness (cf. Simpson—method [8,9]). The sum of these slice volumes yields the ESV and the EDV. Secondarily, the stroke volume (SV) and the ejection fraction (EF) can be calculated from these.

By definition, the papillary muscles are part of the determined areas and thus included in the calculated blood volume, which means that the blood volume is incorrectly overstated in end-diastole and end-systole. This systematic error has no influence on the stroke volume, but the EF is incorrectly calculated as too small. In cases where the EF is a decision criterion for therapy, such as in aortic valve replacement [10] or in the indication for implantation of an intracardiac pacemaker (ICD) [11], an accurate method for determining the EF would be desirable.

The image contrast in the cine-SSFP sequence between the blood and myocardium is sufficient to make a signal-based decision as to whether a blood or myocardial/papillary muscle voxel is being displayed. This should allow for accurate volume information (EDV and ESV) given a known voxel volume.

We implemented this approach in combination with an automatic shape recognition of the heart contour in short-axis slices.

The purpose of this study is to compare a pixel-based evaluation of cardiac function parameters with the classical slice method regarding the difference due to the exclusion of the papillary muscle volume.

## 2. Materials and Methods

The ethics committee of the University Hospital approved the study. All procedures performed in studies with human participants complied with the ethical standards of the institutional research committee and the 1964 Helsinki Declaration and its subsequent amendments or comparable ethical standards. The need for informed consent was waived by the Ethics Committee.

### 2.1. Patients Studied

Cardiac MRI examinations of patients aged 18 years and older between 1 January 2016 and 11 January 2016 were included via our hospital’s Radiology Information System (RIS) for retrospective data collection. Patients with congenital heart defects or cardiac anomalies were excluded. From these patients, 245 were randomly selected alphabetically by first name, and image quality was assessed in the cine short-axis stack. All patients with well-defined cardiac contours in the short-axis cine stack were included in the study [8].

### 2.2. Examination Technique

#### 2.2.1. MR Parameters

All MR examinations were measured on a Magnetom Aera 1.5 T scanner (Siemens Healthineers GmbH, Erlangen, Germany) using an SSFP-CINE sequence with retrospective ECG gating before contrast administration. A body-phased array coil in combination with the spine coil served as the receiver coil. The scan parameters were TR 42.4 ms, TE 1.1 ms and flip angle 55°. The FOV size was 340 × 276 mm, the matrix was 192 × 109 and the slice thickness was 8 mm. This resulted in a voxel size of 1.8 mm × 2.5 mm × 8.0 mm (=36 mm^3^). From the cine sequence, 25 phases were calculated. Depending on the size of the heart, 10–14 layers were measured with a gap of 10% in expiration.

#### 2.2.2. Volume Determination with the Pixel-Based Method (PbM)

The semi-automatic volume determination of the left ventricular heart volume with the pixel-based method (PbM) was evaluated with a software plug-in for the program OsiriX, which was developed by the authors. The software used can be freely purchased from Chimaera GmbH (Erlangen, Germany, www.chimaera.de, accessed on 2 January 2023).

The semi-automatic procedure is based on a brush tool that allows the user to “roughly” colour the target region, such as the left ventricle. In doing so, the algorithm analyses the local image environment of each mouse position and interactively calculates a segmentation mask within the brush size set by the user. The algorithm performs a local intensity analysis based on the minimum, maximum, mean and variance values of the intensity. Using these calculated values, threshold-based region boundaries are determined. Based on each cursor position, local region growth is performed, limited by the determined region boundaries. A morphological “closing” operation additionally closes potential non-detected areas caused by noise in the calculated mask. Optionally, the segmentation tool allows the setting of fixed minima and maxima as intensity thresholds that can be considered in the region’s growth. The software has not been developed specifically for the evaluation of cardiac volumes and can therefore be used for any volume determination if there is sufficient contrast to the surrounding tissue.

For volume determination with the PbM, the cardiac base is first determined in the short-axis layer stack of the end-systole (=smallest subjectively determined area circle in the middle third of the heart) and end-diastole (=first image of the cine sequence). For this purpose, we defined the layer in which the myocardial ring is at least 50% closed as the cardiac base layer [9] (Figure 1).

Starting from this layer, the cavity is marked with the brush tool in each individual layer of the end-systole and end-diastole. For pixel-wise marking, the signal intensity range is selected so that the representation of blood falls within this range. This allows for areas of intraluminal blood to be colour-coded and separated from the myocardium and other surrounding tissues. The resulting areas (Figure 2) can then be added together with the slice thickness to form a volume according to Simpson’s rule [9].

The pixel-based method for evaluating cardiac volumes is a new method where there are no experienced evaluators yet. To enable the most accurate evaluation possible, all data sets were initially evaluated by a non-board certified radiologist. Subsequently, all data sets were checked by a board certified radiologist and modified if necessary.

#### 2.2.3. Volume Determination According to the Contour Surface Method (=KfM) with SyngoVia

All data sets were analysed with syngo.via, version 20A (Siemens Healthineers GmbH, Erlangen, Germany) according to the procedure of Hammon et al. [12]. Here, the software automatically recognizes the cardiac apex and the cardiac base based on the long-axis slices. Along the endocardium and epicardium, the circles are automatically drawn in all heart phases and the volume is calculated. The largest calculated volume is defined as end-diastole and the smallest calculated volume as end-systole. After the automatic segmentation, the evaluator checks the correctness of the heart base and heart apex in end-systole and end-diastole and corrects them if necessary. The circles are then manually checked along the endocardium and epicardium and corrected if necessary. Here, the parameter heart mass is used as an internal control. Since the cardiac mass does not change during the cardiac cycle, the same value should be obtained in end-systole and end-diastole (Figure 3).

All data obtained via this method was internally validated by a board-certified radiologist at our university hospital department.

#### 2.2.4. Volume Determination according to the Contour Area Method (=KfM) with cvi42

All data sets were automatically evaluated using the contour surface method with cvi42 version 5.12 (Circle Cardiovascular Imaging, Montreal, Canada). For this purpose, the Cine short-axis stack and the Cine 2 Ch view were loaded into the program and automatically evaluated with the AI (artificial intelligence) function. The program first automatically determined the systole and diastole and calculated the volumes using the contour area method. The base of the heart was automatically determined in systole and diastole at 2 Ch View. In addition, the papillary muscle volume was segmented automatically. This fact leads to a measured papillary volume with the program cvi42.

#### 2.2.5. Relationship between the Volume Area Method and the Pixel-Based Method

The difference between the two methods is assumed to be the volume of the papillary muscle. This results in the following mathematical correlations:(1)$${ESV}_{KfM}-PM={ESV}_{PbM}$$
(2)$${EDV}_{KfM}-PM={EDV}_{PbM}$$

Equations (1) and (2) show that the stroke volume must be equal for both methods. Thus, the following applies:(3)$$SV={ESV}_{KfM}-{EDV}_{KfM}={ESV}_{PbM}+PM-\left({EDV}_{PbM}+PM\right)={ESV}_{PbM}-{EDV}_{PbM}$$

This does not apply to the ejection fraction (*EF*)
(4)$${EF}_{PbM}=\frac{{EDV}_{PbM}-{ESV}_{PbM}}{{EDV}_{PbM}}=\frac{\left({EDV}_{KfM}-PM\right)-\left({ESV}_{KfM}-PM\right)}{{EDV}_{KfM}-PM}=\frac{{SV}_{KfM}}{{EDV}_{KfM}-PM}$$
or
(5)$${EF}_{KfM}=\frac{{EDV}_{KfM}-{ESV}_{KfM}}{{EDV}_{KfM}}=\frac{\left({EDV}_{PbM}+PM)-({ESV}_{PbM}+PM\right)}{{EDV}_{PbM}+PM}=\frac{{SV}_{PbM}}{{EDV}_{PbM}+PM}$$

From the Formulas (1), (2), (4) and (5), one can now determine the papillary muscle volume in four ways:(6)$$PM1=\frac{{EDV}_{KfM}-{SV}_{KfM}}{{EF}_{PbM}}\;based\;on\;(4)$$
(7)$$PM2=\frac{{SV}_{PbM}}{{EF}_{KfM}-{EDV}_{PbM}}\;based\;on\;(5)$$
(8)$$PM3={EDV}_{KfM}-{EDV}_{PbM}\;based\;on\;(2)$$
(9)$$PM4={ESV}_{KfM}-{ESV}_{PbM}\;based\;on\;(1)$$

We compared the results of the pixel-based method (*PbM*) with the three results of the contour surface method (SyngoVia, cvi42 with papillary muscle, cvi42 without papillary muscle).

The analysis was performed with the programme R (version 3.3.1; open source). All continuous variables (*EDV*, *ESV*, *SV* and *PM*) are given as an average with standard deviation. The volume of the papillary muscles (*PM*) was calculated separately for each of the two contour area methods using the four formulae mentioned above (6)–(9) (e.g., $PM=\frac{SV_{Pbm}}{EF_{KfM}}-EDV_{PbM}$). All results are provided as mean values (minimum; maximum).

The null hypothesis was that papillary muscle volume has no effect on stroke volume or ejection fraction. The null hypothesis was tested using the TOST test for paired samples with a Cohen’s d value of 0.3 and a significance level of 5%.

## 3. Results

A search in our radiology information system yielded 406 adult patients with an MRI cardiac examination between 1 January 2016 and 11 January 2016. For randomization, these were sorted alphabetically by first name and the first 245 were considered in more detail. After reviewing the 245 records, 54 of the 245 patients were excluded due to blurred cardiac contours, e.g., due to respiratory artefacts or arrhythmic heartbeats [9]. Thus, a total of 191 data sets with a gender distribution of 126 (66.3%) male and 65 (33.7%) female participants with a median age of 51 years, and an age distribution at the time of study of 20–75 years could be included in the study.

After the first evaluation of the 191 data sets with the PbM, the stroke volumes (EDV—ESV) were calculated and compared with the stroke volumes of the KfM. In 15 cases, the difference was greater than 15%. After re-evaluation of these data sets with the KfM by an independent investigator who was blinded to the results of the pixel-based method and the first evaluation of the contour area method, five cases remained with a difference in beat volume greater than 15%. In these cases, the pixel-based method (PbM) was re-evaluated. Errors in the program operation, e.g., loading of an incomplete short axis stack when individual layers were repeated due to breathing artefacts—and thus the evaluation of too few layers—could be found as the cause. After a new independent evaluation of these five data sets, the difference in SV was not greater than 15% in any case.

After correcting the datasets, the size of EDV in the pixel-based evaluation was 177 mL (64 mL) [average, (standard deviation)], ESV was 87 mL (62.8 mL), SV was 90 mL (25.5 mL) and EF was 54% (16%). The corresponding values with the gold standard by the cvi42 program were EDV 193 mL (67 mL), ESV 101 mL (66.4 mL), SV 89 mL (25 mL) and EF 50% (14.5%).

The KfM results were EDV 189 mL (66 mL), ESV 100.5 mL (64.26 mL), SV 89 mL (25.4 mL) and EF 50.4% (14.7%). The test for equality of stroke volume between the PbM and the KfM (each with cvi42 and syngo.via) showed equivalence (*p* < 0.001 for TOST upper and TOST lower). However, equivalence cannot be assumed for the comparison of the ejection fractions (TOST lower: *p* < 0.001, TOST upper *p* > 0.001). The EF and SV of the two contour area methods are equivalent (*p* < 0.001 for TOST upper and TOST lower). There was a negative difference for end-diastolic volume and end-systolic volume and a positive difference for ejection fraction. There was no difference in stroke volume (Table 1). The test for equality of EDV and ESV between both KfM methods (cvi42 and syngo.via) showed no equivalence (TOST lower: *p* < 0.001, TOST upper *p* > 0.001). For each part, the upper limit was not significant. Comparing EDV and ESV between the PbM and the KfM method (each with cvi42 and syngo.via) showed no equivalence as well (TOST lower: *p* < 0.001, TOST upper *p* > 0.001).

Figure 4 shows the average left ventricular volumes as a bar diagram. Only, all three values of the stroke volume and the ejection fraction of the clinical finding and gold standard are statistically equivalent.

Using the above formula, an average papillary muscle volume of 14.2 mL (minimum 12.4 mL; maximum 16.3 mL) could be calculated (Table 2).

The automatic segmentation of the papillary muscles using the program cvi42 resulted in an average end-diastolic volume of 5.1 mL and an end-systolic volume of 6.0 mL. The mean of both the end-diastolic and end-systolic volume is exemplary shown in Table 1 (average cvi42: papillary muscle volume).

## 4. Discussion

The measurement of left ventricular function parameters is a common investigation with clinical decision-making relevance. In comparative studies between volume determination using 2D echocardiography and cine MRI, MRI has been shown to be the more accurate method [5,6,7]. The evaluated pixel-based method for ESV and EDV determination is simple and quick to perform and provides an unbiased ESV and EDV as it does not include the papillary muscle volume. The on average 6% higher ejection fraction in PbM compared to KfM can have a significant impact on treatment decisions.

The common practice for volume determination from MR datasets is the contour area method, where the end-diastolic and end-systolic volumes each include the volume of the papillary muscles. In our patient population, the end-diastolic volume (EDV) was on average 6% higher and the end-systolic volume (ESV) was on average 14% higher with the contour area method than with the pixel-based method. The difference can be explained by the fact that in diastole the endocardium is still easily recognizable, but in systole the papillary muscles and endocardium cannot be separated or can only be separated with difficulty (Figure 5), making it more difficult to draw a line along the endocardium.

One approach to solving this problem is to determine the mass of the heart, which is always the same regardless of the cardiac cycle. If the cardiac mass is measured differently in end-diastole and end-systole and the epicardial contour is drawn correctly, a correction of the endocardial cardiac contours is recommended [13,14]. Our data show that despite correction for mass, the difference between the two methods is greater for the ESV than for the EDV. This phenomenon is confirmed by Bailly et al. [15], in whose study the coefficient of variability is larger for the ESV than for the EDV [16,17].

Interestingly, despite a small difference in the average EDV (Δ 5 mL) and ESV (Δ 2 mL), no statistical equality can be assumed when comparing both KfM methods (cvi42 and syngo.via) (cf. Table 1). In particular, for end-diastolic, this could be due to the different proportions of papillary muscle volume within the inner circle.

An important functional parameter for the heart is the ejection fraction (*EF*) = $\frac{SV}{EDV}$. According to Equation (4) ${EF}_{PbM}=\frac{{SV}_{KfM}}{{EDV}_{KfM}-PM}$, the EF is erroneously calculated too low using the contour method, and the deviation depends on the size of the papillary muscles. In our study, the deviation was 6% on average. Riffel et al. [18] compared the cardiac function parameters with and without segmenting out the papillary muscles and confirmed our results. They also found that the stroke volume was higher and EF lower with the contour area method. However, the difference in EF was smaller in Riffel et al., which may have been due to the fact that only healthy subjects were studied.

The ejection fraction is a cardiological parameter that influences the decision on drug or interventional/surgical therapy. For example, an EF ˂ 50% [10] is decisive for the indication of an artificial heart valve for aortic valve replacement. The EF is also a relevant parameter in the therapy decision trees for treatment concepts of other heart valve diseases, e.g., mitral valve insufficiency [19].

In patients with heart failure, the ejection fraction is not only a diagnostic parameter but also an important functional parameter for therapy decisions concerning both drug therapy and the implantation of a left ventricular assist device [20]. In dilatative cardiomyopathy, implantation of an ICD has indicated if the chronic left ventricular ejection fraction is less than 35% [11]. Furthermore, EF ≤ 40% is included in the CHADS-VASC score as a criterion for heart failure. Thus, it indirectly contributes to the consideration of stroke risk and the resulting decision to use oral anticoagulation for patients with atrial fibrillation [21,22].

Many evaluators of cardiac volumes refer to the standard values published in 2015 in JCMR by Kawel-Böhm et al. [23]. The “standard values” given here for men and women up to and from 60 years of age were determined with the KfM, but the papillary muscles were segmented separately and not evaluated as part of the LV volume. Thus, referencing the measurement results with the KfM to these standard values [23] is severely limited. Likewise, a re-evaluation of the application to current norm values is necessary before the clinically practical use of PbM. Further studies are needed, especially to set new reference values. We would like to emphasize that the gender ratio in our study is asymmetrical (66% male and 34% female). This aspect should be considered in future studies, especially for the setting of new reference values.

The difference between the two methods is the volume of the papillary muscles. According to the formula for calculating the volume, the volume in our study is 14.2 mL on average. The automatic segmentation of the papillary muscles resulted in a value of 5.1 mL end-diastolic and 6.0 mL end-systolic and is clearly smaller than determined via the pixel-based method. The reason for this is that the cvi42 program only segments the papillary muscle located in the lumen, but not the part that is directly adjacent to the myocardium. This also explains the difference between systolic and diastolic volumes of the automatically determined papillary muscle. This results in a larger proportion of the papillary muscle volume for smaller left ventricular volumes. As expected, PbM results in lower end-diastolic and end-systolic volumes compared to KfM. Consecutively, the ejection fraction becomes larger with PbM. The measurement of end-diastolic and end-systolic volume with inclusion of the papillary muscles should therefore be questioned.

The pixel-based method solves the “papillary muscle problem” in a technically simple way by automatic signal detection within predefined limits in combination with contour detection. It is not a “specialist” for cardiac evaluation and can determine many other volumes where there are good signal or density differences to the surrounding tissue. Despite using a development version of the software without automation algorithms, the heart volume of the left ventricle could be determined in about 2 min. A complete automation of the method is conceivable in the future.

In the future, more and more artificial intelligence (AI) will be used in every aspect of our daily lives, including medical applications. He et al. compared the left ventricular ejection fraction echocardiographically using a sonographer vs. artificial intelligence [24]. They were able to show that AI is not inferior to the measurement of the left ventricular ejection fraction with echocardiography compared to a sonographer. Future applications could therefore include AI systems to assist examiners in imaging problems.

A limitation of our study is the unknown actual stroke volume. With an ECG-triggered flow measurement in the ascending aorta, which is not routinely measured in our department, this could have been determined, as flow measurement provides more reproducible results than volumetry [25,26]. Another limitation is that the volume of anatomical structures with bizarre morphology, such as the papillary muscles can only be determined with estimates based on various assumptions. A solution would be direct measurements in body donors and comparison with post-mortem MRI, as already successfully measured by Bertozzi et al. [27].

Before establishing it as a standard, another limitation is the exclusion of patients with moderate image quality and consequently insufficiently identifiable heart contours. This seemed to be in most cases with atrial fibrillation while detecting a higher heart rate during the assessment.

## 5. Conclusions

PbM is easy and quick to perform for the determination of left ventricular heart function. In terms of stroke volume, it provides comparable results to the established disc/contour area method and measures the actual left ventricular heart function, leaving out the papillary muscles. This results in an average 6% higher ejection fraction, which can have a significant influence on therapy decisions.

## Figures and Tables

**Figure 1 diagnostics-13-01437-f001:**
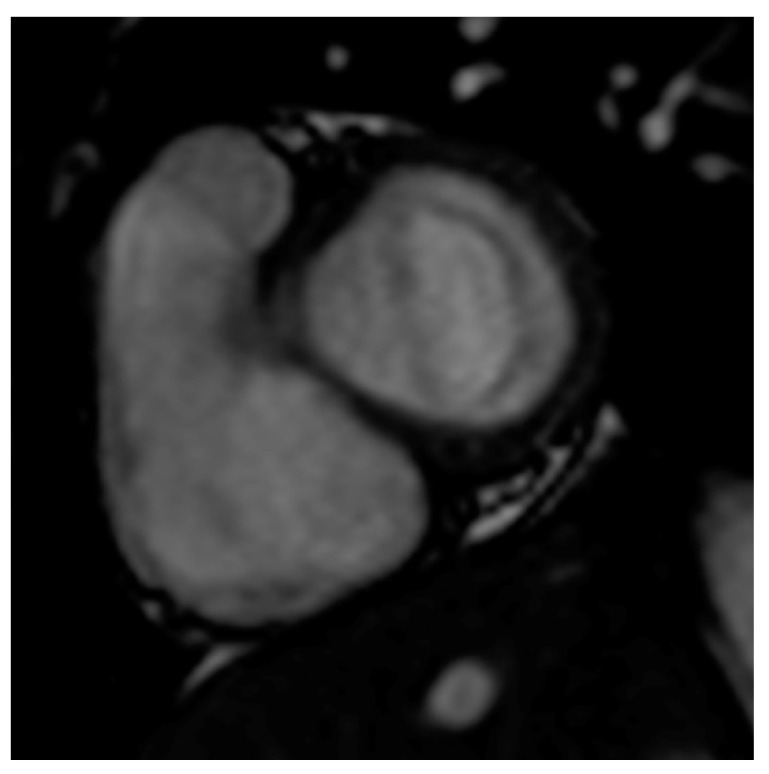
Determination of the most basal layer of the left ventricle at the transition to the left atrium. The myocardial ring is at least 50% complete.

**Figure 2 diagnostics-13-01437-f002:**
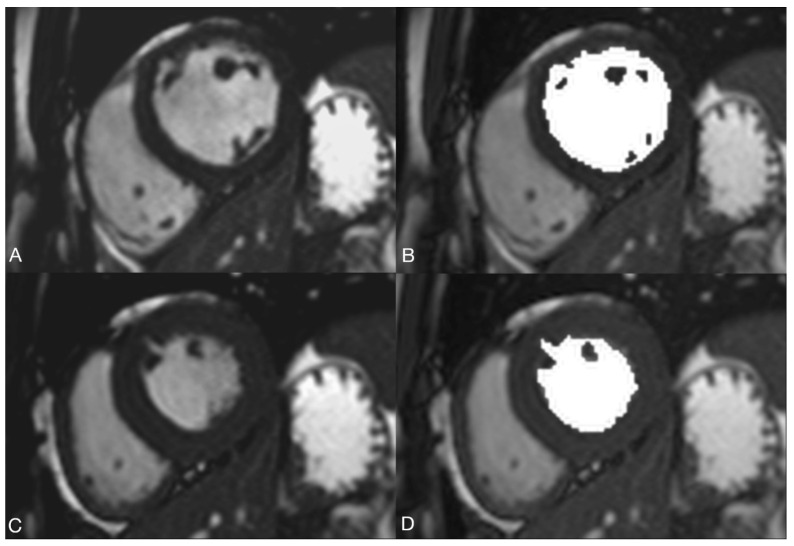
Evaluation of an exemplary layer in diastole (**A**,**B**) and systole (**C**,**D**) with the pixel-based method.

**Figure 3 diagnostics-13-01437-f003:**
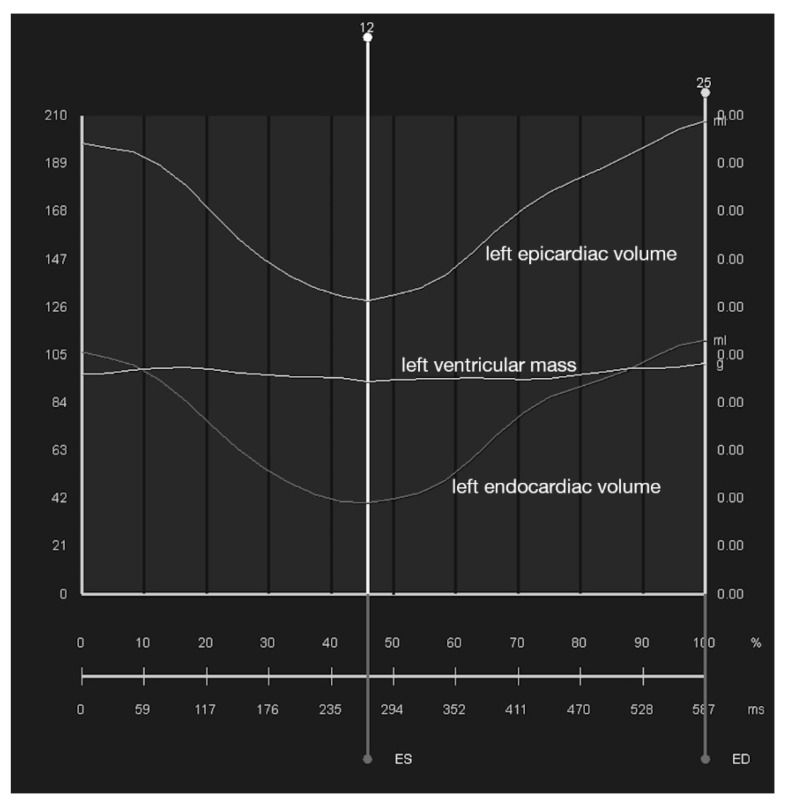
Evaluation of epicardial and endocardial volume and cardiac mass over the entire cardiac cycle. Ideally, the mass line is horizontal over the entire cardiac cycle.

**Figure 4 diagnostics-13-01437-f004:**
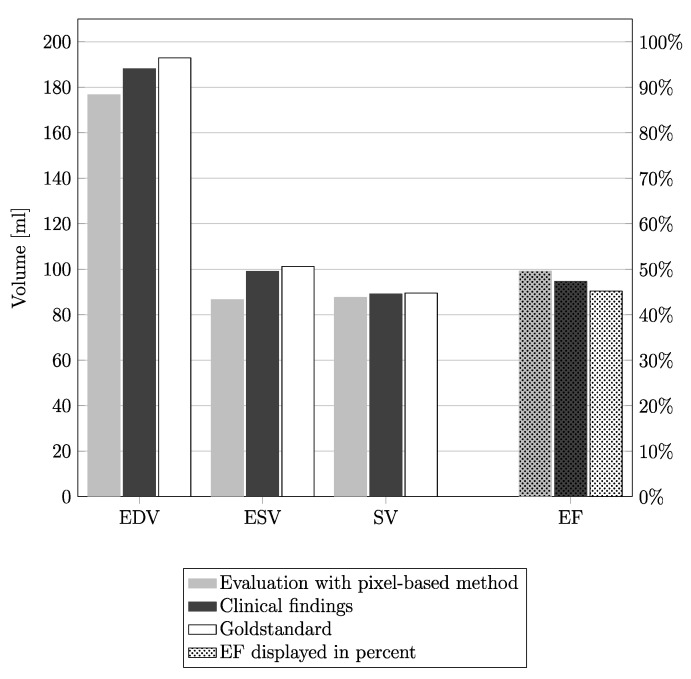
Mean values of the parameters EDV (=end-diastolic volume), ESV (=end-systolic volume), SV (=stroke volume) and EF (=ejection fraction) of all datasets (*n* = 208) using the pixel-based method (grey columns), the conventional contour surface method (black columns) and the gold standard (white columns). *y*-axis: volume [mL].

**Figure 5 diagnostics-13-01437-f005:**
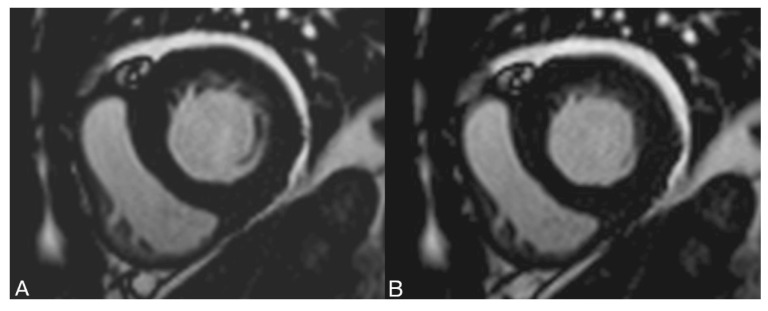
Good visual differentiation of the papillary muscle and the endocardium in the presystolic phase (**A**) and lack of border between papillary muscle and endocardium in systole (**B**).

**Table 1 diagnostics-13-01437-t001:** Mean values: EDV = end-diastolic volume, ESV = end-systolic volume, SV = stroke volume, EF = ejection fraction, PMV = papillary muscle volume, PbM = pixel-based evaluation method, KfM = contour surface method, cvi42 = evaluation method with cvi42 program (gold standard), * calculated value, ** mean papillary muscle volume given by the program cvi42, *** values in this line are statistically equivalent according to Tost-Test.

	Average PbM	Average KfM	Average cvi42
EDV [mL]	177	188	193
ESV [mL]	87	99	101
SV [mL]	88 ***	89 ***	90 ***
EF [%]	50	47 ***	45 ***
PMV [mL]	—	14.2 *	5.5 **

**Table 2 diagnostics-13-01437-t002:** Calculated, average papillary muscle volumes.

Formula	(6)	(7)	(8)	(9)	Mean
KfM syngo.via [mL]	13.4	15.4	12.4	13.5	13.6
KfM cvi42 [mL]	15.7	16.3	13.9	13.3	14.8
Mean [mL]	14.5	15.8	13.1	13.4	14.2

## Data Availability

The datasets analyzed during the current study are available from the corresponding author on reasonable request.

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
