# Peer review of "Cardiac MRI: An Alternative Method to Determine the Left Ventricular Function"

_diagnostics, 2023, doi:10.3390/diagnostics13081437_

Round 1

Reviewer 1 Report

After examining the scientific study, the following considerations may be made. The scientific study is well structured in all its parts. In particular, the premises with which the authors introduced the analysis are clear. Equally clear are the objectives that led the authors to carry out this study and the section on materials and methods. Particular appreciation can also be expressed for the material on which the study was carried out. The data was collected methodically and without bias. The results were consistent and significant and allowed a discussion section full of food for thought. The authors then developed a discussion of the results achieved.

The number and quality of the citations are appropriate; however, the scientific relevance of the article could benefit from an expansion of the same. In the specific advice to add the following quotes:

  • Bertozzi G, Cafarelli FP, Ferrara M, Di Fazio N, Guglielmi G, Cipolloni L, Manetti F, La Russa R, Fineschi V. Sudden Cardiac Death and Ex-Situ Post-Mortem Cardiac Magnetic Resonance Imaging: A Morphological Study Based on Diagnostic Correlation Methodology. Diagnostics (Basel). 2022 Jan 17;12(1):218. doi: 10.3390/diagnostics12010218. PMID: 35054385; PMCID: PMC8774558.

Author Response

We would like to thank the reviewers for contributing to the improvement of the paper through their meaningful and supportive comments.   

We would like to say thank you for the new work that helps us to improve the number and quality of quotes. The recommended article was a good suggestion. We added it in the discussion section. 

Furthermore, we added a new study published on the 5th of April 2023. He et al compared the left ventricular ejection fraction echocardiographically measured by a sonographer vs. artificial intelligence to highlight further possibilities in imaging in the future. 

(He, B., Kwan, A.C., Cho, J.H. et al. Blinded, randomized trial of sonographer versus AI cardiac function assessment. Nature (2023). https://doi.org/10.1038/s41586-023-05947-3) 

Reviewer 2 Report

Hessman  et al compared the papillary muscle inclusion vs exclusion methods in terms of overall LV ejection fraction, stroke volume and other LV volumes. The conclusion was that PbM provides higher cardiac function, which could be clinically significant. 

This is an important study addressing a fairly common clinical question. Sample size and data analytical approaches are adequate.

Comments-

1. The patient selection 66% male and 34% female - is asymmetric given that this manuscript is aiming to define reference cardiac anatomic measurements. 

2.  Cross-comparison of the data outputs generated by experienced vs inexperienced investigators is an unusual approach. 

3. No inter-modality validation. The observation that PbM provides "true LV function" and higher EF is somewhat arbitrary.

4. Computational methods are well-described. Statistical analyses are missing for the most part. 

Author Response

We would like to thank the reviewers for contributing to the improvement of the paper through their meaningful and supportive comments.  

  1. As you pointed out there is an asymmetric gender selection. To reveal new standard references, we now pointed out in our work that further studies taking gender equality into account are required. (line 298-301)
  2. To ensure that our data is not just good clinical practice, we decided to achieve high standards. Therefore, we chose to first collect the data from a non-board certified radiologist who is well versed in the operation of the new program for evaluation with the pixel-based method and have the results reviewed by an experienced radiologist. We have written this section in more detail. (line 124-127)

  3. We do not know the "true" volume of the papillary muscles and thus do not know the "true" LV function. We wanted to approach the "true" volume of the papillary muscles by determining the "true" EDV and ESV with the pixel based method. The difference between the EDV and ESV in our new method and the contour area method should be the volume of the papillary muscle. (line 267-273 and 330-334)

  4. Our aim was to verify the data statistically by using the test of equivalence (TOST-Test). We added a statistical part for EDV and ESV to our materials and methods section. We discuss it further in the revised version (see section results and discussion). Each value in table one is now statistically proven. In table two we marked the calculated values as calculated; those are mathematical proven as described in the methodical part. (line 216-221)

Reviewer 3 Report

The reviewer would like to thank the authors for the opportunity to review their work.

The study discusses the measurement of left ventricular function parameters in cardiac imaging, focusing on the differences between two methods: the pixel-based method (PbM) and the contour area method.

The PbM is shown to be simpler, quicker, and more accurate than the contour area method, which includes the volume of papillary muscles and can lead to inaccurate ejection fraction (EF) calculations.

The difference between the two methods is the volume of the papillary muscles, and automatic segmentation of the papillary muscles may not be accurate in measuring the volume.

I only have 2 comments: 

First in Line 76 : "...245 were randomly selected..". Could you , plese, provide more details about the randomisation process? (sequential? random number generator? etc.

The second is about the line 285: "According to the formula for calculating the volume, the difference in our study is 14.2 mL on average." which contradicts with table 1 and 2.- could you, please, clarify this?.

Author Response

We would like to thank the reviewers for contributing to the improvement of the paper through their meaningful and supportive comments.  

As mentioned in the results we randomly selected our data alphabetically by first name. To clarify the randomization process we decided to explain it earlier in the paper. The process is now detailed in the section ‘material and methods’. 

We are also grateful for your second comment. To clarify this, we changed the word ‘difference’ to ‘volume’ in line 285, as our calculation results in a volume (14.2 ml).

Round 2

Reviewer 1 Report

After examining the scientific study, the following considerations may be made. The scientific study is well structured in all its parts. In particular, the premises with which the authors introduced the analysis are clear. Equally clear are the objectives that led the authors to carry out this study and the section on materials and methods. Particular appreciation can also be expressed for the material on which the study was carried out. The data was collected methodically and without bias. The results were consistent and significant and allowed a discussion section full of food for thought. The authors then developed a discussion of the results achieved.